# Dissecting the molecular evolution of fluoroquinolone-resistant *Shigella sonnei*

Hao Chung The [1], Christine Boinett[1,2], Duy Pham Thanh[1], Claire Jenkins[3], Francois-Xavier Weill [4], Benjamin P. Howden [5], Mary Valcanis[5], Niall De Lappe[6], Martin Cormican[7], Sonam Wangchuk[8], Ladaporn Bodhidatta[9], Carl J. Mason [9], To Nguyen Thi Nguyen[1], Tuyen Ha Thanh[1], Vinh Phat Voong[1], Vu Thuy Duong[1], Phu Huong Lan Nguyen[1,10], Paul Turner [2,11], Ryan Wick [12], Pieter-Jan Ceyssens[13], Guy Thwaites[1,2], Kathryn E. Holt [12,14], Nicholas R. Thomson[14,15], Maia A. Rabaa [1,2,17]* & Stephen Baker [1,2,16,17]

*Shigella sonnei* increasingly dominates the international epidemiological landscape of shigellosis. Treatment options for *S. sonnei* are dwindling due to resistance to several key antimicrobials, including the fluoroquinolones. Here we analyse nearly 400 *S. sonnei* whole genome sequences from both endemic and non-endemic regions to delineate the evolutionary history of the recently emergent fluoroquinolone-resistant *S. sonnei*. We reaffirm that extant resistant organisms belong to a single clonal expansion event. Our results indicate that sequential accumulation of defining mutations (*gyrA*-S83L, *parC*-S80I, and *gyrA*-D87G) led to the emergence of the fluoroquinolone-resistant *S. sonnei* population around 2007 in South Asia. This clone was then transmitted globally, resulting in establishments in Southeast Asia and Europe. Mutation analysis suggests that the clone became dominant through enhanced adaptation to oxidative stress. Experimental evolution reveals that under fluoroquinolone exposure in vitro, resistant *S. sonnei* develops further intolerance to the antimicrobial while the susceptible counterpart fails to attain complete resistance.

[1] The Hospital for Tropical Diseases, Wellcome Trust Major Overseas Programme, Oxford University Clinical Research Unit, Ho Chi Minh City, Vietnam. [2] Centre for Tropical Medicine and Global Health, Oxford University, Oxford, UK. [3] Gastrointestinal Bacterial Reference Unit, National Infection Service, Public Health England, London, UK. [4] Institut Pasteur, Unité des Bactéries Pathogènes Entériques, Paris, France. [5] Microbiological Diagnostic Unit Public Health Laboratory, Department of Microbiology and Immunology, Peter Doherty Institute for Infection and Immunity, The University of Melbourne, Melbourne, Australia. [6] National Salmonella, Shigella, and Listeria monocytogenes Reference Laboratory, University Hospital Galway, Galway, Ireland. [7] School of Medicine, National University of Ireland Galway, Galway, Ireland. [8] Public Health Laboratory, Department of Public Health, Ministry of Health, Royal Government of Bhutan, Thimphu, Bhutan. [9] Department of Enteric Diseases, Armed Forces Research Institute of Medical Sciences, Bangkok, Thailand. [10] The Hospital for Tropical Diseases, Ho Chi Minh City, Vietnam. [11] Cambodia-Oxford Medical Research Unit, Angkor Hospital for Children, Siem Reap, Cambodia. [12] Department of Infectious Diseases, Central Clinical School, Monash University, Melbourne, VIC 3004, Australia. [13] Unit Bacterial Diseases, Sciensano, Brussels, Belgium. [14] London School of Hygiene and Tropical Medicine, London, UK. [15] The Wellcome Trust Sanger Institute, Hinxton, Cambridge, UK. [16] Cambridge Institute of Therapeutic Immunology and Infectious Disease, The Department of Medicine, University of Cambridge, Cambridge, UK. [17] These authors jointly supervised this work: Maia A. Rabaa, Stephen Baker. *email: mrabaa@oucru.org

$S$higella is ranked among the leading causes of diarrhoeal disease in children under five years of age[1,2], and causes ~160,000 deaths per year across all age groups globally. The Shigella genus is comprised of four serogroups (S. dysenteriae, S. boydii, S. flexneri and S. sonnei); however, the majority of Shigella infections are associated with organisms belonging to S. flexneri and S. sonnei. S. flexneri has traditionally been associated with disease in developing countries, while S. sonnei has been associated with more developed settings. This trend, however, is changing, with S. sonnei prevailing in modernising countries, particularly in Asia[3,4].

The principal virulence determinant of Shigella is the type III secretion system encoded by a large virulence plasmid, which enables the pathogen to invade the colonic mucosa and kill resident macrophages, triggering inflammatory responses and causing damage to the intestinal epithelium[5]. It is this damage that causes the distinctive symptom of bloody/mucoid diarrhoea in shigellosis. Although the majority of Shigella infections are self-limiting, shigellosis can be severe if left unattended[6]. Routine treatment relies on antimicrobials to accelerate recovery, prevent complications, and reduce onward transmission. Current World Health Organization (WHO) guidelines recommend the use of ciprofloxacin, a fluoroquinolone (FQ), for the treatment of shigellosis[6]. Fluoroquinolones are a safe, well tolerated, highly effective antimicrobial class with broad-spectrum activity and are highly efficacious for treating Shigella-associated diarrhoea. However, the widespread use of fluoroquinolones has led to increasing resistance in many Shigella species, raising questions about their utility[7]. Indeed, FQ-resistant (FQr) Shigella have been placed on the global priority list for pathogens requiring urgent antimicrobial development[8]. The U.S. Centers for Disease Control and Prevention (CDC) now recommend that fluoroquinolones should not be prescribed for infections caused by Shigella with a minimum inhibitory concentration (MIC) of >0.12 μg/mL against ciprofloxacin (https://emergency.cdc.gov/han/han00401.asp). This threshold is considerably lower than the currently defined 1 μg/mL breakpoint for ciprofloxacin resistance defined by the European Committee on Antimicrobial Susceptibility Testing (EUCAST) (http://www.eucast.org/clinical_breakpoints/).

We recently demonstrated that the extant FQr S. sonnei population, now reported worldwide, likely emerged from a single common ancestor in South Asia, the descendants of which moved across Asia prior to intercontinental dissemination[9]. Despite the significance of these findings, our inferences were derived from limited sampling, consisting of genome sequences generated from 70 FQr S. sonnei isolates. Here, we include >300 globally sourced FQr S. sonnei to understand the life history of this looming public health threat, particularly within the phylogenetic context provided by contemporary South Asian isolates. This extensive collection (~400 organisms over a timespan of 20 years) allows us to delineate how this FQr S. sonnei clone evolved both within and outside of South Asia, thus revealing in detail its evolutionary history, global epidemiology, pangenome repertoire, and adaptive signatures. Additionally, we further evaluate the evolutionary trajectories of FQr and FQ-susceptible (FQs) S. sonnei under experimental conditions, showing that the two variants exhibit differing patterns of mutation accumulation.

## Results

**The clonal expansion of FQr S. sonnei.** Our previous analysis of 70 representative FQr S. sonnei suggested that they emerged as a single clone (nested within CenAsiaIII) in South Asia and subsequently radiated internationally[9]. Here, we utilised a collection of 411 internationally isolated S. sonnei, including 265

contemporary isolates sequenced for the purpose of this study (Table 1). Hypothesising that South Asia was the most likely origin of FQr S. sonnei, isolates were preferentially selected from collaborating institutes if they originated from this region or showed reduced susceptibility to ciprofloxacin (intermediate to full resistance). This approach resulted in the over-representation of South Asian S. sonnei (306/411; from India, Bhutan, Nepal, Bangladesh, Pakistan, and Sri Lanka), spanning a period of 16 years (1999–2014). Within this collection, 76% (313/411) of the isolates exhibited phenotypic resistance to ciprofloxacin (defined as FQr) and were isolated as early as 2008. These emphases allowed us to reconstruct the evolutionary history of FQr S. sonnei with fine granularity. Preliminary phylogenetic reconstruction showed that 11 isolates clustered outside the CenAsiaIII clade, and these were removed from further phylogenetic analyses; five of these exhibited reduced susceptibility to ciprofloxacin (MIC ranges 0.094–0.125 μg/mL).

Maximum likelihood (ML) phylogenetic reconstruction of 395 CenAsiaIII S. sonnei verified that almost all FQr organisms (98.7%; 307/311) belonged to a single clone, with FQr conferred by the classical sequential mutations in the quinolone resistance determining region (QRDR): gyrA-S83L, parC-S80I and gyrA-D87G (Fig. 1). These data confirmed that the current global burden of FQr S. sonnei was strongly associated with a single clonal expansion event. We observed a single FQr isolate (Sh_sonnei_1446) that was a sister taxon to the major FQr clone, sharing with it the gyrA-S83L and parC-S80I mutations, but harbouring a distinct gyrA codon 87 mutation (gyrA-D87N). This observation highlights the presence of other non-sampled FQr S. sonnei isolates with differing secondary gyrA mutations, although these appeared to be present at lower frequencies. In addition, QRDR single mutants, each carrying a mutation in either gyrA codon 83 or 87, appear to have arisen on at least six independent occasions, leading to an initial reduced susceptibility to FQ (i.e. incomplete resistance). Plasmid-mediated quinolone resistance (PMQR) genes were uncommon, with qnrS1 found in two S. sonnei (already harbouring the aforementioned triple mutation) and no organisms possessing aac(6′)-lb-cr, qepA, or oxqAB.

Bayesian hierarchical clustering using core SNPs segregated the CenAsiaIII clade into two distinct subpopulations, which we refer to herein as Pop1 (mainly FQs; $n = 83$) and Pop2 (mainly FQr; $n = 312$) (Supplementary Fig. 1). To reconstruct the evolutionary history of the CenAsiaIII clade, we selected 72 isolates from each of these two subpopulations and subjected them to Bayesian phylogenetic analysis using BEAST. The mean nucleotide substitution rate for the entire CenAsiaIII clade was estimated to be $7.34 \times 10^{-7}$ substitutions per site per year [95% highest posterior density (HPD): $6.77 \times 10^{-7}$ to $7.91 \times 10^{-7}$], and we infer that this clade likely emerged in the late 1980s (95% HPD: 1986–1990) (Fig. 2). This timeline is in accordance with previous estimates from an analysis of the global evolutionary history of S. sonnei[10]. The tree topology shows that Pop2 emerged from within Pop1 (Fig. 2), and allowed us to estimate the timing of stepwise QRDR mutation accumulation leading to the formation of the major FQr clone.

Within Pop1, two lineage-defining gyrA mutations likely arose independently within the mid-1990s, forming two major subclades with quinolone (i.e. nalidixic acid) resistance: gyrA-S83L in ~1995 (95% HPD: 1994.6–1997) and gyrA-D87Y in ~1996 (95% HPD: 1995.4–1998) (Fig. 2). The gyrA-S83L lineage later gave rise to Pop2 and its dominant FQr subclade, following the acquisition of parC-S80I between 2001 (95% HPD: 2000.5–2002.6) and 2005 (95% HPD: 2003.9–2006.4), and then gyrA-D87G around 2007 (95% HPD: 2006.5–2007.6) (Fig. 2). Bayesian phylogenetic analysis of the Pop2 subclade yielded similar estimates in terms of the timeline of emergence of the

**Table 1 Summary of sequences used to investigate the emergence of fluoroquinolone-resistant *Shigella sonnei***

| Country | Number of sequences | In CenAsiaIII clade | CIP resistant isolates | Study or Institute of origin | Patient group | Region of recent travel history (N) | Sequencing platform | CIP susceptibility test |
|---|---|---|---|---|---|---|---|---|
| Bhutan | 71 | 71 | 71 | Diarrhoeal disease surveillance in JDWNRH, Thimphu, Bhutan (AFRIMS) | Hospitalised children <5 years old | NA | Illumina HiSeq 2000 | Disk diffusion/E-test |
| Vietnam | 24 | 22 (~2 poor sequencing quality) | 22 | Diarrhoeal disease surveillance in HCMC, Vietnam | Hospitalised children <5 years old | NA | Illumina MiSeq | Disk diffusion |
| Thailand | 8 | 8 | 1 | Cross-sectional study of S. sonnei diversity in Southeast Asia (AFRIMS) | Hospitalised children <5 years old | NA | Illumina HiSeq 2000 | Disk diffusion |
| Cambodia | 1 | 1 | 1 | Cross-sectional study of S. sonnei diversity in Southeast Asia (AFRIMS) | Hospitalised children <5 years old | NA | Illumina HiSeq 2000 | Disk diffusion |
| Ireland | 20 | 20 | 16 | National *Salmonella, Shigella* and *L. monocytogenes* Reference Laboratory, Galway, Ireland | Primarily patients with recent travel history | South Asia (9), Africa (1), Europe (1), No travel (5), Unknown (4) | Illumina HiSeq 2000 | Broth microdilution |
| Australia | 85 | 77 | 46 | Microbiological Diagnostic Unit Public Health Laboratory, Melbourne, Australia | Primarily patients with recent travel history | Africa (2), America (1), Southeast Asia (12), Middle East (2), Oceania (1), South Asia (60), Unknown (4) | Illumina NextSeq | Agar dilution |
| France | 97 | 95 | 72 | French National Reference Centre for *E. coli, Shigella*, and *Salmonella*, Paris, France | Primarily patients with recent travel history | Africa (4) America (1) Southeast Asia (4) South Asia (88) | Illumina HiSeq 2000 | Disk diffusion |
| England | 91 | 90 | 82 | Gastrointestinal Bacteria Reference Unit, Public Health England, London, UK | Primarily patients with recent travel history | Africa (3), East Asia (1), Europe (13), Middle East (1), South Asia (73) | Illumina HiSeq 2000 | Disk diffusion |
| Global references | 14 | 11 | 0 | NA | NA | NA | Illumina HiSeq 2000 | NA |
| Total | 411 | 395 | 311 | | | | | |

*CIP* ciprofloxacin, *JDWNRH* Jigme Dorji Wangchuk National Reference Hospital, *AFRIMS* Armed Forces Research Institute of Medical Sciences, *HCMC* Ho Chi Minh City

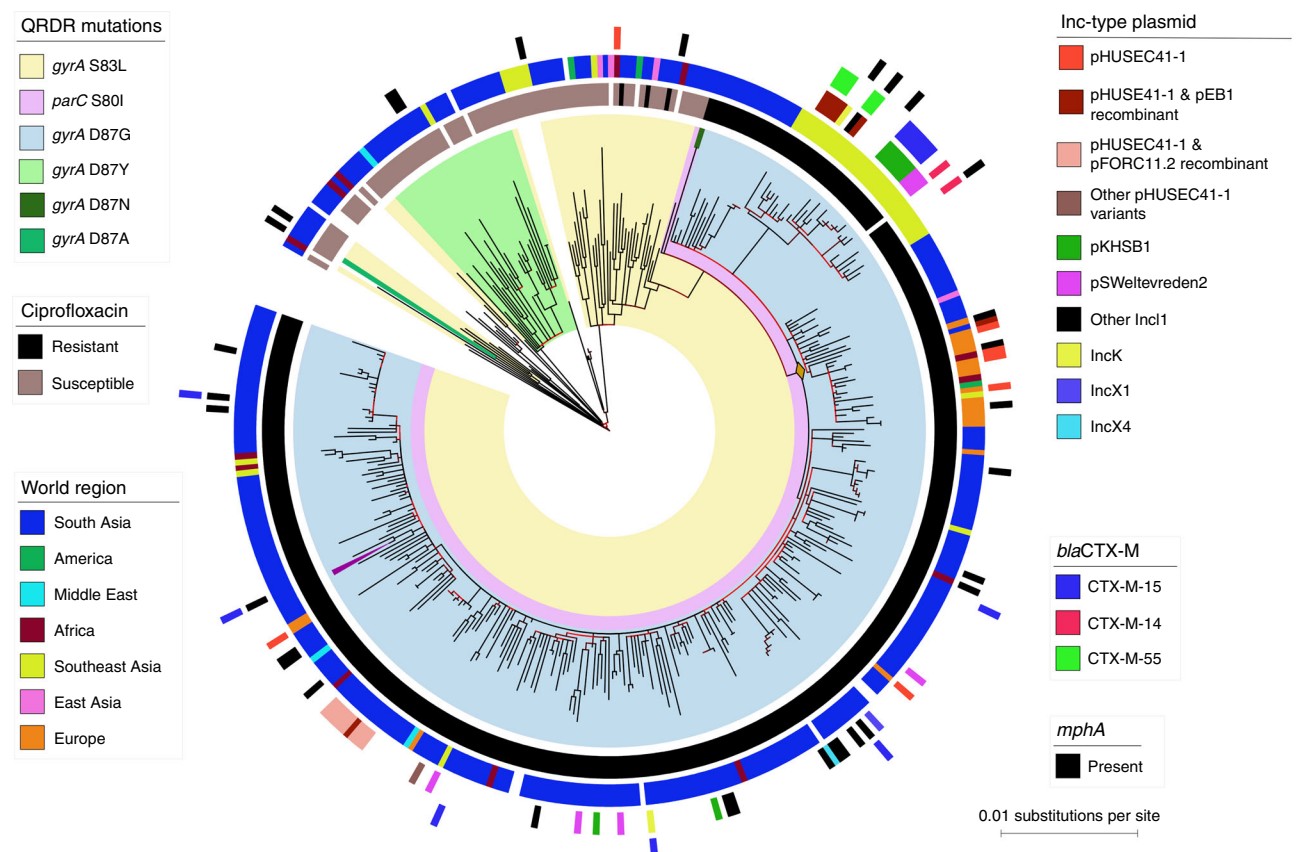

**Fig. 1** The phylogenetic structure of CenAsiaIII *Shigella sonnei*. The figure displays the maximum likelihood phylogeny of 395 *S. sonnei* sequences belonging to the CenAsiaIII clade. The tree is rooted on the most closely related sequence from the *S. sonnei* global sequencing study. The branch colour scheme indicates bootstrap support for the corresponding branch, ranging from low to high (red to black). The shading covered on the phylogeny indicates taxa harbouring quinolone resistance determining region (QRDR) mutations (see key). The orange diamond indicates the internal node leading to the major fluoroquinolone-resistant (FQr) clone. The magenta triangle denotes the position of taxon 2012–02037, which was subjected to long-read sequencing (see Methods). The rings show the associated information for each taxon, from the innermost to the outermost in the following order: (1) susceptibility testing for ciprofloxacin (black: resistance; brown: susceptible); (2) original geographical source; (3) presence of different plasmids; (4) presence of ESBL *bla*CTX-M genes; (5) presence of *mphA*. The horizontal scale bar indicates the number of nucleotide substitutions per site

major FQr clone. We separately conducted a BEAST analysis of only this clone to examine its demographic history, inferring that its population expanded rapidly after emergence in 2007, peaked around 2008, and maintained a relatively constant size thereafter (Supplementary Fig. 2).

**Geographical dissemination of FQr *S. sonnei* from South Asia.** It is apparent in the ML phylogeny of all CenAsiaIII isolates that South Asian isolates are diverse and deep branching across the tree, consistent with an origin in this region (Fig. 1). To test this quantitatively, we generated 1000 geographically down-sampled phylogenies (from the aforementioned ML phylogeny) to reduce sampling location bias (with equal representation of South Asia, Europe, Africa, and Southeast/East Asia) and employed stochastic mapping to evaluate the most probable geographical origin of the CenAsiaIII clade (see Methods). These analyses supported South Asia as the primary hub for intercontinental spread, with CenAsiaIII *S. sonnei* being transmitted from this region to Africa, Southeast/East Asia, and Europe on an average of 7.65 (IQR: 5–9), 3.47 (IQR: 2–5), and 3.8 (IQR: 3–4) independent events per tree, respectively (summarised across the 966 subsampled phylogenies with successful stochastic mapping) (Fig. 3a). In addition, approximately half of the evolutionary time inferred across the entire CenAsiaIII phylogeny was predicted to be in South Asia, providing further evidence that the clade has been

circulating and evolving principally in this region (Fig. 3b). Analyses of tip-location randomised phylogenies produced significantly different inferences compared to those estimated from the true ML phylogeny (Supplementary Fig. 3). In particular, the evolutionary time spent in South Asia was significantly higher in the true ML phylogeny, compared to all randomised datasets ($p < 0.05$ in 10/10 comparisons, ANOVA-Tukey's test [F value: 186.3, df: 10]). These results provide strong evidence that our conclusions concerning the South Asian origin of the clade result from inherent phylogeographic signal in the data, rather than over-representation of South Asian isolates in this collection.

Two distinct regional diversification events from South Asia are evident within the FQr clone, with one being introduced into Southeast Asia (Vietnam, Thailand, and Cambodia; light yellow box in Fig. 2) by 2010 (95% HPD: 2010–2011.7), and another instigating sustained transmission within Europe (Ireland, Italy, Germany, and Spain; light orange box in Fig. 2) from 2009 (95% HPD: 2009–2010.2). Notably, despite a dense sampling strategy, there was no immediate outgroup for the Southeast Asia subclone (Fig. 2). Detailed examination identified 18 chromosomally dispersed mutations specific to this subclone, confirming its divergence from the trunk of the phylogeny. We speculate that this long branch may be the result of a recent population bottleneck or intermediate unsampled isolates that were vital to transition from South Asia to Southeast Asia. Within this subclone was a number of organisms isolated from Australian

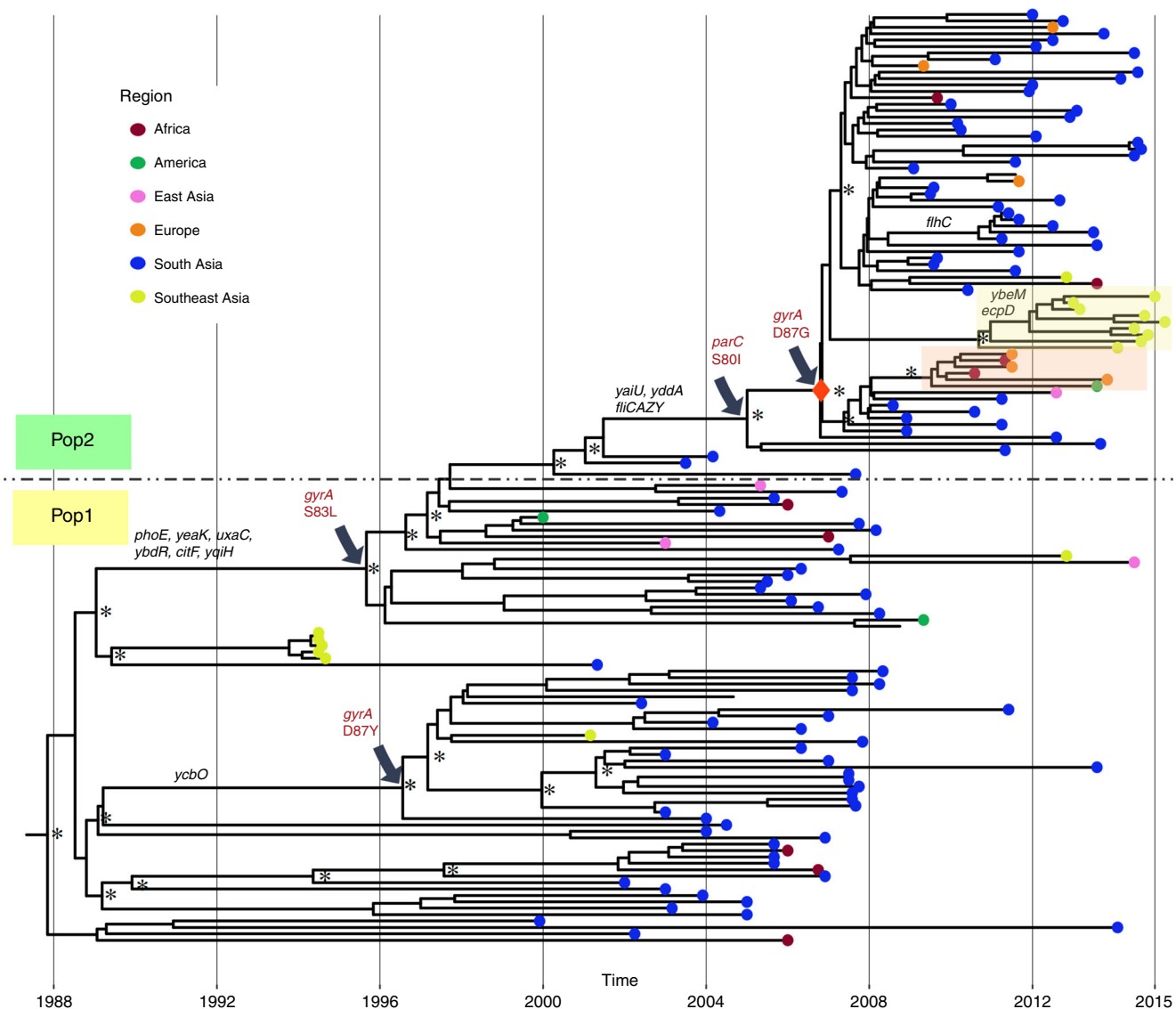

**Fig. 2** Temporal phylogenetic reconstruction of CenAsiaIII *Shigella sonnei* overlaid with pseudogenes. A maximum clade credibility phylogenetic reconstruction of 144 CenAsiaIII *S. sonnei* is shown. Asterisks indicate posterior probability support >80% on major internal nodes. The red diamond indicates the internal node leading to the major fluoroquinolone-resistant (FQr) clone. Grey arrows designate branches on which certain quinolone resistance determining region (QRDR) mutations occurred. Lineage-defining pseudogenes and gene loss events are indicated above corresponding branches. Tip colours show the original geographical isolation regions of taxa (see key). The light orange and light yellow boxes highlight the European and Southeast Asian clonal expansions of CenAsiaIII, respectively. The dark dashed line separates the two populations (Pop1 and Pop2) in CenAsiaIII, defined by Bayesian hierarchical clustering

and European patients (12 occasions), thus highlighting the association between international travel and secondary introductions to shigellosis non-endemic populations. The second subclone, comprised mainly of organisms from Europe, shared its most closely related outgroups with sequences from South Asia, again suggesting likely introduction from this region (Figs. 1 and 2).

**Multi-drug resistance in FQr *S. sonnei*.** Aside from FQ resistance, it was important to characterise the entire antimicrobial resistance gene (ARG) content of organisms within the CenAsiaIII population. Genes encoding resistance to previous first-line antimicrobials for treating shigellosis were commonplace within the clade, as observed previously[10,11]. These included the small spA plasmid, conferring resistance to streptomycin (*strAB*),

tetracycline (*tetRA*), and sulphonamide (*sul2*), and a class II integron conferring resistance to trimethoprim (*dfrA1*). Other notable ARGs included those belonging to the extended spectrum beta-lactamase (ESBL) *bla*$_{\text{CTX-M}}$ family; these were particularly enriched in the Southeast Asia subclone (Fig.1), but only sporadically detected among the South Asia isolates. Three *bla*$_{\text{CTX-M}}$ variants were carried on different plasmid structures; these were acquired independently in Southeast Asia, specifically Vietnam (Fig. 1, Supplementary Fig. 1). While *bla*$_{\text{CTX-M-55}}$ mainly co-existed with an IncB-O plasmid (recombinant of pEB1 and pHUSEC41-1) in five Vietnamese isolates, the *bla*$_{\text{CTX-M-15}}$ variant was co-transferred with the IncI1 plasmid pKHSB1 and clonally maintained in six additional Vietnamese isolates. Notably, pKHSB1 and its *bla*$_{\text{CTX-M-15}}$ have been the adaptive signature of the indigenous Global III-clade *S. sonnei* population in Ho Chi Minh City since 2006[12]. This observation highlights the

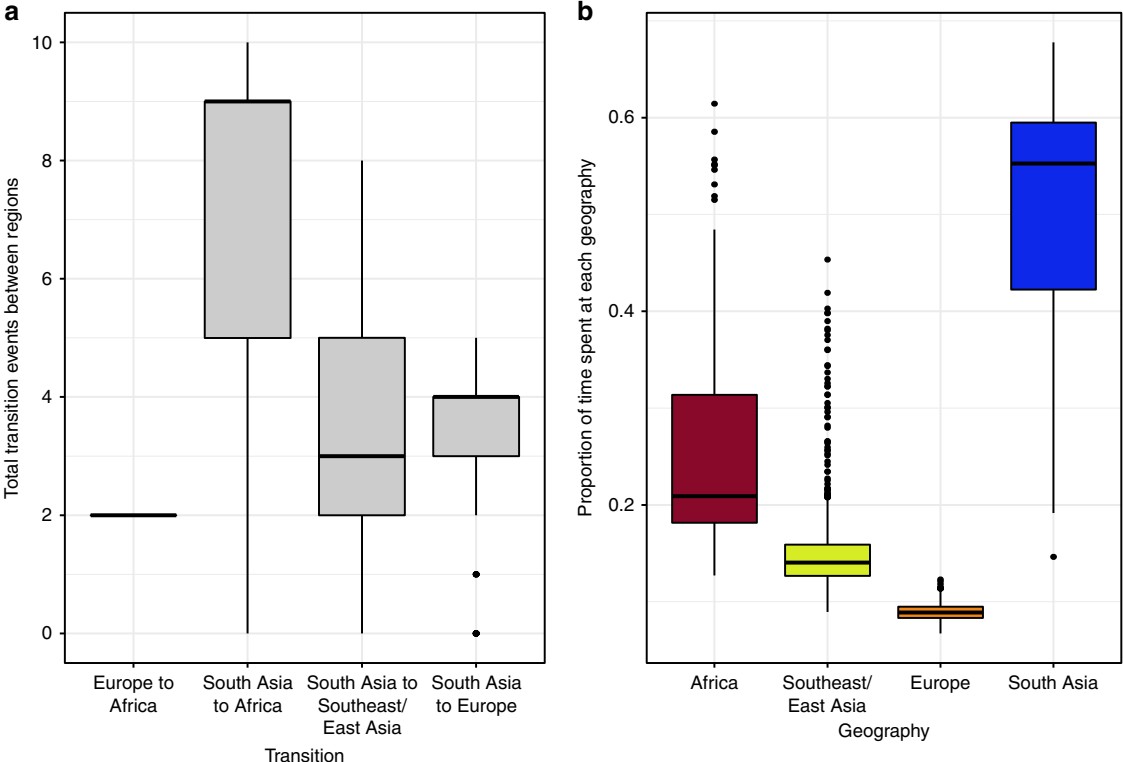

**Fig. 3** Phylogeographical inference of CenAsiaIII *Shigella sonnei*. Stochastic mapping was used to estimate **a** the number of transitions between different geographical states: Africa, Southeast/East Asia, Europe, and South Asia, with transitions between states that are less than one considered non-significant and not represented in the plot, and **b** the proportion of time spent in each geographical state. In both panels, boxplots summarise results of 966 successful subsamplings of the phylogeny described in Fig. 1, and stochastic mapping was performed for each with 100 simulations

promiscuity of this specific plasmid background among different *S. sonnei* lineages circulating in the same location. We additionally recorded four independent instances of *mphA* (conferring azithromycin resistance) acquisition within this Vietnamese expansion, two of which were co-transferred with *bla*$_{CTX-M}$ genes (Fig. 1), and have characterised the genetic details of these pan-resistant *S. sonnei* elsewhere[13]. Significantly, this combination of mutations and ARG renders these organisms resistant to all current first-line antimicrobials used for dysentery treatment (FQ, 3rd generation cephalosporins, and macrolides).

We catalogued the presence of major plasmids in all CenAsiaIII *S. sonnei* and found that IncB-O and IncI1 plasmids were the most common (Fig. 1). Detailed manual plasmid comparisons revealed that all IncB-O plasmids were closely related to pHUSEC41-1 or recombinant variants (nucleotide identity >98%, coverage >80%). These plasmids exhibited associations with specific geographical locations (Supplementary Data 2). Specifically, pHUSEC41-1 was found to circulate primarily in Europe, while the pHUSEC41-1-pEB1 and pHUSEC41-1-pFORC11.2 recombinants were found in Vietnam and Bhutan, respectively (Fig. 1, Supplementary Fig. 1). Notably, plasmids of the pHUSEC41-1 family were favourably maintained in the FQr *S. sonnei* following local introductions, despite geographical discrepancy. The IncI1 plasmids, present in 42 *S. sonnei* isolates, displayed greater diversity than the IncB-O plasmids with respect to their origins. These IncI1 plasmid structures were similar to IncI1 plasmid structures found in GenBank for *Salmonella* spp. (*n* = 14 plasmids), *E. coli* (*n* = 13), *S. sonnei* (*n* = 11), and *K. pneumoniae* (*n* = 3) (Supplementary Data 2). Most notably, two Vietnamese subpopulations separately harboured pKHSB1 (*S. sonnei*) and pSWeltevreden2 (originally described in *Salmonella* Weltevreden) (Fig. 1, Supplementary

Fig. 1). These data demonstrate that once FQr *S. sonnei* enters a new location, it has access to a wide gene pool shared among the indigenous Enterobacteriaceae. In addition, we previously reported that CenAsiaIII *S. sonnei* harbours a novel recombinant colicin plasmid (pSSE3)[11]. Through genomic comparison and Bayesian phylogenetic inference, we confirmed that this plasmid has been stably maintained in this clade since its introduction around the year 1995, concurrent with the first *gyrA* S83L mutation (Supplementary Fig. 1). Plasmid pSSE3 likely grants CenAsiaIII *S. sonnei* a competitive advantage over other *Shigella* and *E. coli* through colicin-mediated killing activity[14]. Furthermore, various clusters of different colicin families (*colE2, colE4, colE5, colE8, colIa, colIb, colJs, colK,* and *colD*) have been sporadically acquired across the phylogeny, with *colIa* and *colIb* integrated into the co-transferred IncI1 plasmid backbones on at least 13 occasions (Supplementary Fig. 1). These data show that maintenance and enrichment of colicin clusters may also facilitate *S. sonnei* in forming a successful clonal expansion.

**Reductive evolution in CenAsiaIII S. sonnei.** Reductive evolution has played an important role in shaping the long-term evolution of *Shigella*[15]. However, this phenomenon has not been sufficiently studied in a rapidly emerging population. Understanding gene loss and pseudogenization may provide insight into the evolutionary trajectory of this CenAsiaIII clade. We selected one isolate (2012–02037; highlighted in Fig. 1) for long-read Nanopore sequencing and complete genome assembly to aid in the detection of pseudogenes arising from IS disruptions (See Methods). Our analyses identified a total of 27 pseudogenization events stably inherited across the phylogeny (9 IS disruptions and 18 by other mechanisms; Supplementary Table 1). Specifically,

the *S. sonnei* population had accumulated 14 pseudogenes prior to the formation of CenAsiaIII. Since the emergence of CenAsiaIII in the late 1980s, 13 additional lineage-defining pseudogenizations have arisen (Fig. 2). All of these mutations were located in genes encoding transporters (*araH*, arabinose; *ycbO*, aliphatic sulfonate; and other substrate nonspecific transporters *yjcR*, *phoE*, and *yddA*) and oxidation-reduction processes (*ydhV*, *ybeQ*, *mdaA*, and *ybdR*). Additionally, we identified an inactivation in *nfsA*, which has been described to confer resistance to nitrofurans[16]. Additional observed mutations potentially disrupt metabolic pathways, such as glucuronate (*uxaC*) and citrate (*citF*) metabolism. A truncation in *yeaK*, encoding a deacylase capable of correcting tRNAs mischarged with serine/threonine (ProXp-ST1), has been shown to negatively impact translation fidelity[17]. However, tolerable levels of mistranslation have been shown to increase fitness in *E. coli* subjected to oxidative stress[18].

The most distinguishing mutation was the obliteration of the flagellin monomer and its transcriptional unit, *fliCAZY*; we speculate this occurred in a stepwise fashion, beginning with the insertion of an IS630 element into *fliC* at a specific target sequence (5′-CTAG-3′) prior to the divergence of CenAsiaIII[19]. This IS element, in combination with an IS630 present downstream of *fliY*, may act as an excision hotspot, leading to the deletion of the entire operon. Pan-genome analysis demonstrated that deletion of this region has occurred on multiple occasions, including one occurring prior to the formation of the FQr population (Pop2) in 2005. The *fliC* gene encodes flagellin, a highly immunogenic protein and the monomer for flagellum assembly, while *fliA* encodes for the flagellum-specific sigma factor (Eσ[28]). Though *S. sonnei* is conventionally non-motile, flagellum production has been observed, albeit at a low magnitude compared to other Enterobacteriaceae[20]. The removal of the entire *fliCAZY* operon leads to irreversible loss of the whole flagellar assembly. This observation further explains the subsequent inactivation of the redundant flagellar synthesis master regulator, *flhC*, in a Bhutanese *S. sonnei* subclone (Fig. 2). Furthermore, the loss of *fliA* in *E. coli* led to better survival during stress via increased energy conservation and physical membrane stabilisation, which resulted from the loss of the motor proteins and the flagellum apparatus[21]. A previous study also found that non-flagellated *P. putida* possesses greater endurance under oxidative stress due to the release of more energy and NADPH, which is otherwise coupled to the flagellar machinery[22].

**Mutation analysis of CenAsiaIII *S. sonnei*.** Recurrent mutations in the same gene in differing organisms serve as a hallmark of convergent evolution, which may highlight a common adaptive strategy to a specific selective pressure[23]. We aimed to detect potential signals of convergent evolution by examining the incidence of both pseudogenization and missense mutations occurring within the evolutionary history of CenAsiaIII. Our analysis revealed that *rpoS* is the most frequent target for inactivation (46 occasions) and missense mutations (23 occasions). However, such mutations in *rpoS* may also be laboratory artefacts due to long-term storage[24]. Other genes that showed evidence of convergent pseudogenization included *pepT* (14 occasions), *ynfK* (8 occasions), and *puuD* (7 occasions) (Supplementary Table 2). Notably, *pepT* encodes a tripeptidase, which is critical for the degradation of glutathione, a principal buffer for oxidative stress[25]. Consequently, the inactivation of this gene increases cellular glutathione concentration, potentially granting these organisms a higher redox capability. Similarly, *puuD* encodes an intermediate enzyme responsible for the conversion of putrescine into GABA. Inactivation of *puuD*, combined with the pre-existing *puuE* pseudogene, can preferentially reroute putrescine to the production of spermidine, which scavenges free radicals and heightens resistance to oxidative stress[26]. Alternatively, *ynfK* encodes a putative dethiobiotin synthetase, catalysing the penultimate step in biotin biosynthesis. We did not, however, observe inactivating mutations in its paralogue *bioD*, implying that biotin biosynthesis is more likely to be downregulated rather than inactivated. In *E. coli*/*Shigella*, biotin acts as the key co-factor for the acyl-CoA carboxylases (ACCs) to convert acetyl-CoA into fatty acid. Upon host cell invasion, *Shigella* imports fatty acids from the host cytosol. Therefore, depletion in biotin may direct *Shigella*'s acetyl-CoA pool away from fatty acid production and conserve it for energy generation via the conversion of acetyl-CoA into acetate[27]. These examples show that though the aforementioned mutations are not fixed in the population, their multiple independent occurrences may provide a transient fitness advantage, thus enhancing the progression of infection within a human host.

Selection analysis identified genes with a high density of non-synonymous mutations, which is a measure of possible positive selection. Among the 3637 *S. sonnei* homologues examined, 14 genes were predicted to have undergone positive selection (adjusted dN/dS ranging from 1.8 to 3.6) (Table 2). Six of these candidate genes were antimicrobial targets (*gyrA* and *parC*) or

**Table 2 Genes predicted to be under selection in the CenAsiaIII *Shigella sonnei* population**

| Gene | No. of N[a] SNPs | No. of S[b] SNPs | Adjusted dN/dS | Product | GO[c] process |
|---|---|---|---|---|---|
| *mreB* | 10 | 0 | Inf | Dynamic cytoskeleton protein | Regulation of cell shape |
| *gyrA* | 8 | 0 | Inf | DNA gyrase subunit A | Response to antibiotic |
| *putA* | 8 | 1 | 3.557 | Proline dehydrogenase | Response to oxidative stress |
| *sigA* | 7 | 0 | Inf | Serine protease | NA |
| *ygjU* | 6 | 0 | Inf | Serine/threonine symporter | Amino acid transport |
| *ylbE* | 5 | 0 | Inf | Hypothetical protein | NA |
| *ynfK* | 5 | 0 | Inf | Dethiobiotin synthetase | Biotin biosynthesis |
| *ubiH* | 5 | 0 | Inf | 2-octaprenyl-6-methoxyphenol hydroxylase | Ubiquinone biosynthesis; response to oxidative stress |
| *parC* | 5 | 0 | Inf | DNA topoisomerase IV subunit A | Response to antibiotic |
| *yahG* | 5 | 1 | 2.22 | Membrane | NA |
| *yieO* | 5 | 1 | 2.223 | Putative transport protein MFS[d] | Transmembrane transport |
| *SSON1290* | 4 | 1 | 1.778 | Hypothetical protein | NA |
| *ydbK* | 4 | 1 | 1.778 | Pyruvate flavodoxin oxidoreductase | Response to oxidative stress |
| *hyfB* | 4 | 1 | 1.778 | Hydrogenase 4, component B | Oxidation-reduction |

[a]Non-synonymous single-nucleotide polymorphisms
[b]Synonymous single-nucleotide polymorphisms
[c]Gene ontology biological process, as defined in EcoCyc for *Escherichia coli* K-12 MG1655
[d]Major facilitator superfamily

mediators of cellular oxidation-reduction processes (*putA*, *ubiH*, *ydbK*, and *hyfB*). Several of these genes (*mreB*, *gyrA*, *ubiH*, and *hyf*) were previously identified as undergoing diversifying selection in the resident Vietnamese *S. sonnei* Global III population[12], confirming that adaptation to oxidative stress is a common adaptive strategy for this pathogen. Notably, *ynfK*, which was found to be regularly inactivated, may also be under selection, demonstrating that partial cessation of biotin biosynthesis is favoured by *S. sonnei*.

**Experimental evolution reveals differential mutational profiles between FQr and FQs *S. sonnei*.** Our recent diarrhoeal surveillance study performed in HCMC, Vietnam during 2014–2016 found that CenAsiaIII FQr *S. sonnei* has gradually replaced the resident FQs Vietnamese *S. sonnei* population[28]. We conducted an in vitro evolution experiment aiming to identify differing mutational profiles in these two distinct lineages upon exposure to (fluoro)quinolones. A summary of the mutations arising during this experiment are described in Table 3. For the ciprofloxacin-susceptible strain cipS-VN (*gyrA*-S83L; Global III), exposure to either nalidixic acid or ciprofloxacin for ~450 generations increased the MIC against both nalidixic acid (>256 µg/mL) and ciprofloxacin. The endpoint MICs for ciprofloxacin varied between experimental conditions and replicates, but did not reach the defined breakpoint for resistance (1 µg/mL) (Table 3). Exposure to nalidixic acid in cipS-VN consistently led to disruptive mutations in the transcriptional repressor, *emrR*. Such mutations are known to result in overexpression of the EmrAB pump, also heightening resistance to nalidixic acid[29]. Other explanatory mutations included a SNP in *parE* and a disruptive mutation in *acrR*. Mutations in *parE* were commonly identified when cipS-VN organisms were exposed to ciprofloxacin, either as a non-synonymous SNP or a deletion stably

inherited in both replicate populations. In addition, exposure to ciprofloxacin, but not nalidixic acid, triggered a stable and specific genomic deletion (*treF-yhjA-yhi-hde-slp-iucABCD-iutA*). This region spans genes encoding cytochrome C peroxidase (*yhjA*), acid stress resistance (*yhi* and *hde* loci), and aerobactin biosynthesis (*iuc* biosynthesis locus and *iutA* receptor). This region in *S. sonnei* is flanked by an IS4 transposase, which may explain the mechanism of genomic deletion. It has been recently proposed that bactericidal antimicrobials, including fluoroquinolones, can induce cell death via the production of reactive oxygen species (ROS)[30]. The common penultimate step in this process is the stimulation of the Fenton reaction, which recruits hydrogen peroxide and ferrous ions to produce highly cytotoxic hydroxyl radicals. Consequently, high concentrations of intracellular iron facilitate rapid FQ-mediated killing. Our data suggest that the cipS-VN *S. sonnei* circumvents this by lowering intracellular iron via the obliteration of the aerobactin siderophore. Additionally, mutations targeting *cysE* were consistently observed across all culture conditions for cipS-VN, but not for cipR-VN. *cysE* encodes a serine acetyltransferase, catalysing the conversion of L-serine to the O-acetyl-L-serine (OAS) quorum sensing signal. Previous studies have reported that *cysE* mutants exhibit an improved capacity for in vitro biofilm formation through the suppression of OAS production[31]. Additionally, minimal medium, as implemented here, is known to induce biofilm formation in *E. coli*[32]. Therefore, consistent *cysE* mutations may be a general response to selective growth in minimal medium.

Exposure of cipR-VN (*gyrA*-S83L, *parC*-S80I, and *gyrA*-D87G; CenAsiaIII) to nalidixic acid similarly led to enrichment of *emrR* mutants, which in one case was associated with an elevated MIC to ciprofloxacin from 8 to 12 µg/mL. Long-term exposure to ciprofloxacin doubled the MIC to ≥16 µg/mL, and both experimental replicates (CIP-A and CIP-B) showed mutations in *rob*. The *rob* gene encodes the transcription activator Rob, a

**Table 3 Results of antimicrobial susceptibility and mutation analysis of experimental evolution in *Shigella sonnei***

| Condition[a] | MIC-CIP (µg/mL) | MIC-NAL (µg/mL) | Resistance mutations | Other mutations[d] | Genomic deletion |
|---|---|---|---|---|---|
| cipS-VN: D0[b] | 0.25 | 64 (sat) | | | |
| cipS-VN: M9-A | 0.19 | 64 (sat) | | *cysE* (P252S) | |
| cipS-VN: M9-B | 0.25 | 64 (sat) | | *cysE* (P252T); *clpA* (T374I); *metJ* (D76N, A61E); *rpoS* (Q161P) | |
| cipS-VN: NAL-A | 0.5 | >256 | ***emrR***[c] (fs); ***parE*** (P571S) | *cysE* (A33E); *cysE* (S249R); *cysB* (P145L) | |
| cipS-VN: NAL-B | 0.38 | >256 | *acrR* (fs); *emrR* (fs) | *cysE* (Q228K); *dinG* (fs, 2 a.a deletion); *metJ* (R78P, A13T) | |
| cipS-VN: CIP-A | 0.38 | >256 | ***parE*** (S458L) | *cysE* (fs); *cysE* (P252T) | *treF-yhjA-yhi-hde-slp-iucABCD-iutA* |
| cipS-VN: CIP-B | 0.75 | >256 | ***parE*** (2 a.a insertion) | ***cysE*** (M256I) | *treF-yhjA-yhi-hde-slp-iucABCD-iutA* |
| cipR-VN: D0 | 8 | >256 | | | |
| cipR-VN: M9-A | 8 | NA | | ***rpoS*** (R277L); ***iucA*** (stop codon) | |
| cipR-VN: M9-B | 6 | NA | | ***rpoS*** (R301H); *iucD* (L113P); *ycjC* (P47S); SSON_2732 (2 a.a insertion) | |
| cipR-VN: NAL-A | 12 | NA | *emrR* (fs) | SSON_1790 (stop codon); ***hemX*** (fs) | |
| cipR-VN: NAL-B | 8 | NA | ***emrR*** (I27 deletion) | **SSON_2588** (A93T) | |
| cipR-VN: CIP-A | 16 (sat) | NA | ***rob*** (G245R) | *eutQ* (fs) | |
| cipR-VN: CIP-B | 16 (sat) | NA | *rob* (Q194K); *emrR* (fs) | SSON_1804 (N200K) | |

*fs* frameshift mutation
[a]In all conditions, selected strains were cultured in minimal M9 medium supplemented with glucose and niacin (25 µg/mL) with the addition of: (1) M9-A/B: no antimicrobial, (2) NAL-A/B: the same concentration of nalidixic acid (256 µg/mL). (3) CIP-A/B: ciprofloxacin in concentrations of half the corresponding MIC of the respective strain (0.125 µg/mL for cipS-VN and 4 µg/mL for cipR-VN). cipS-VN indicates the use of a ciprofloxacin-sensitive *S. sonnei* (*gyrA*-S83L; Global III clade) strain and cipR-VN indicates the use of a ciprofloxacin-resistant *S. sonnei* (*gyrA*-S83L, *gyrA*-D87G and *parC*-S80I; CenAsiaIII clade) strain. (sat): satellite colonies grown inside the zone of inhibition.
[b]D0: initial condition on day 0.
[c]Mutations in bold indicate those that appear to be fixed in the examined population.
[d]Excluding synonymous mutations, and mutations found in intergenic regions and repetitive genetic elements.

protein composed of two functional domains. The N-terminal domain shares substantial amino acid homology to SoxS (responsive to oxidative stress) and MarA (responsive to antimicrobials), indicating that the Rob regulon overlaps with those defined for these two common regulators[33]. It has been proposed that the C-terminal domain represses the activity of Rob by sequestration, impeding its access to the transcriptional machinery[34]. In our experiment, both missense mutations observed in *rob* fell into its C-terminal coding region, potentially changing its conformation and releasing more active Rob. Overexpression of Rob is associated with resistance to multiple antimicrobials, including quinolones[35], owing to its increased activation of multi-drug efflux pumps (directly or indirectly via *marRAB*) and decreased expression of the outer membrane porin, *ompF* (via upregulated *micF*)[36,37]. Our examination on the mutational profiles of 395 CenAsiaIII *S. sonnei* revealed that there was only one non-synonymous mutation in *emrR* and no mutation in *rob*. In the absence of antimicrobials, cipR-VN maintained fixed mutations in *rpoS*, both of which fell into the defined DNA binding residues of sigma38 ($E\sigma^S$)'s domains 4.1 (R277L) and 4.2 (R301H)[38]. This region mediates $E\sigma^S$ binding to the −35 promoter region of regulated genes. However, most promoters recognised by $E\sigma^S$ show poor conservation in the −35 element, and mutagenesis of $E\sigma^S$ domain 4 shows an insubstantial effect on promoter binding strength and specificity[39,40].

## Discussion

The large and diverse collection utilised in this study enhanced our capacity to investigate the origins and international dissemination of FQr *S. sonnei*. We confirmed that global FQr *S. sonnei* are clonal and likely emerged in South Asia around early 2007. This finding is in agreement with a wealth of epidemiological data citing the rise of FQr *S. sonnei* since 2007[9]. It is notable that the second and third mutations (*parC*-S80I, *gyrA*-D87G) were acquired rapidly in the last two decades, coinciding with a sharp increase in the use of FQ during this period, when these drugs became routinely prescribed for shigellosis[6,41]. We provide evidence of FQr *S. sonnei*'s sustained circulation in Southeast Asia and Europe, but the extent of its propagation is likely to be underestimated in regions with limited representation in our analysis, such as East Asia, Africa, and the Americas. Indeed, the same clone was confirmed to have caused a recent (2015) outbreak in California[42]. Furthermore, FQr *S. sonnei* harbouring identical QRDR mutations are prevalent among the community of men who have sex with men (MSM) in Taiwan[43]. These outbreaks suggest that the organism is spreading rapidly worldwide and poses a significant threat to public health. Transmission within the MSM network should be closely monitored, especially in the wake of increased shigellosis incidence and antimicrobial resistance within this high-risk group[44–46]. Although we report here that the major resistance clone has accumulated triple QRDR mutations, we could not exclude the existence of other resistance mechanisms. A year-long surveillance study of *S. sonnei* in England and Wales identified seven FQr organisms with the combinatorial mutations *gyrA*-S83L, *parC*-S80I, and *gyrA*-D87N, which are identical to a single resistant isolate in our collection[47]. Resistance due to a synergy of the PMQR gene *qnrB* and *gyrA* mutations has also been noted in India[48]. The frequent gains of resistance determinants to other first-line treatments, ESBL and *mphA*, further exacerbate the problem. This issue is emphasised in Southeast Asia, where the gene pool of the local Enterobacteriaceae is likely to be vast and serves as a substantial reservoir of resistance genes. We surmise that the propagation of acquired resistance elements is likely to be underrepresented in Europe and South Asia, as these organisms were not sourced during active surveillance. Indeed, FQr *S. sonnei* circulating among the MSM community in England and Australia have recently acquired the epidemic *mphA*-carrying plasmid pKSR100, demonstrating that differing antimicrobal exposures and transmission dynamics in different patient groups may select for distinguished AMR patterns in this novel pathogen variant[46,49].

Our genomic examination highlighted the prevalence of reductive evolution mechanisms in CenAsiaIII *S. sonnei*, which arose over a thirty-year timeframe. Although it is not possible to fully catalogue pseudogenes generated via IS element disruptions, we observed some noticeable evolutionary trends. These included the disruption of several metabolic functions, specifically the fermentative processes of alternate carbon sources, as well as the cessation of cellular appendages such as flagella and pili. Furthermore, our analysis pointed to selection for better adaptation under oxidative stress. This is supported by the finding that most of the genes identified to be subject to convergent evolution and potential selection are linked to oxidation-reduction processes. We speculate that the high pressure of oxidative stress may originate from varied sources but is most likely due to the oxidative environment within the macrophage or exposure to antimicrobials such as fluoroquinolones. Improved survival during stress could benefit FQr *S. sonnei* clones in their expansion into different geographical destinations. However, this interpretation may be subject to scrutiny given that dN/dS selection analyses are not particularly well-suited for investigating rapidly evolving intraspecific bacterial populations[50]. Specifically, the nucleotide differences observed within a population do not represent fixed divergences but rather transient polymorphisms, which are not necessarily purged from the population due to the short evolutionary timeframe considered in this study.

Exposure to a quinolone antimicrobial resulted in disruption of the efflux pump repressor, EmrR, for both FQr and FQs *S. sonnei* despite having different genetic backgrounds. In addition, we showed that mutations in the Rob transcriptional activator result in resistance to FQ at very high MIC. It is surprising that, despite its harbouring of a *gyrA* mutation, continuous exposure of this FQ-susceptible *S. sonnei* to ciprofloxacin failed to generate specific resistance mutations in *gyrA* and *parC*. Previous in vitro studies have also reported that in *E. coli* with single *gyrA* mutation background, mutations linked to drug efflux but not specific QRDR targets are preferentially selected[51]. This finding also corroborates the observation that indigenous Vietnamese *S. sonnei* did not accumulate triple mutations to achieve FQ resistance, even though fluoroquinolones are commonly used in this setting. FQ resistance attributed to QRDR mutations has been shown to not be associated with detrimental fitness cost in Enterobacteriaceae in vitro, even in the absence of FQ[52,53]. Here we predict that, in the absence of quinolone pressure, FQr *S. sonnei* showed mutational bias toward alternating the *rpoS*-induced stress response. Since the experimental evolution component only included two replicates per strain-condition combination, a larger study may be performed to confirm whether *rpoS* mutations are truly compensatory for the resistance genotype.

Though we elucidated the evolution and spread of this emerging FQr *S. sonnei* population, the long-term clinical and public health significance of this clone remains unclear. Our recent observations in Vietnam demonstrated that disease severity and recovery in *S. sonnei*-infected children were independent of the organism's ciprofloxacin susceptibility profile[28]. Additionally, evolutionary trajectories and development of resistance to other antimicrobial classes in already FQr isolates in different ecological niches require further investigation. Future research should be focused on the collateral issues associated with FQ and to explore how FQr affects other patient cohorts, including the malnourished, the elderly, and the immunocompromised.

## Methods

**Organism collection and whole-genome sequencing**. This investigation centring on the international dissemination of FQr *S. sonnei* was performed by combining WGS data from our previous investigations with an additional 265 isolates sequenced for this study[9,11]. This resulted in a collection of 411 *S. sonnei* isolates (Bhutan, $n = 71$; Vietnam, $n = 24$; Thailand, $n = 8$; Cambodia, $n = 1$; Ireland, $n = 20$; Australia, $n = 85$; France, $n = 97$; England, $n = 91$; global references, $n = 14$). Table 1 provides a summary of the organisms used in this study; associated metadata are detailed in Supplementary Data 1. The isolates from shigellosis endemic countries (Bhutan, Vietnam, Thailand, Cambodia) are mostly FQr and sourced from multiple diarrhoeal surveillance studies in the respective countries. Institutional review board (IRB) approval and informed consent for these studies have been described previously[9]. Isolates from shigellosis non-endemic regions were collected from public health laboratories (Ireland, Australia, England, France), and these were mostly isolated from adults with recent travel histories. These isolates were collected from anonymous sources at local public health laboratories, and were not subject to IRB approval, so informed consent was not required. When available, the patients' recent travel records were reported to identify the original geographic sources of the infections. Isolates were selected preferentially if they showed reduced ciprofloxacin susceptibility (intermediate to full resistance to ciprofloxacin), or if they originated from South Asia. Susceptibility to ciprofloxacin was determined using a variety of methods, including disk diffusion, E-test, agar dilution, and broth microdilution, as detailed in Table 1. The breakpoint for ciprofloxacin resistance was inferred following guidelines from the European Committee on Antimicrobial Susceptibility Testing (http://www.eucast.org/clinical_breakpoints/)[9]. Specifically, MIC to ciprofloxacin >1 μg/mL is classified as being resistant. For all *S. sonnei* isolates, genomic DNA was extracted using commercial kits and subjected to WGS on various Illumina platforms to produce paired-end short-read sequences (Table 1). These sequence data were then consolidated for analysis and interpretation.

**Short read mapping and phylogenetic reconstruction**. All reads were mapped to the *S. sonnei* Ss046 reference sequence (accession number: NC_007384) using SMALT (version 0.7.4). High-quality SNPs were called and filtered using SAMtools and bcftools, removing those matching any of the following criteria: consensus quality < 50, mapping quality < 30, ratio of SNPs to reads < 75%, read depth < 4, and number of reads per strand < 2. We previously found that all sequenced FQr *S. sonnei* clustered within the Central Asian expansion of Lineage III (CenAsiaIII clade)[9]. Preliminary phylogenetic reconstruction demonstrated that among 411 examined sequences, only eleven were not of the CenAsiaIII clade. These eleven sequences, together with another two sequences with poor mapping quality and three most closely related sequences as outgroups to the CenAsiaIII clade, were removed to create a final set of 395 sequences belonging to this clade and utilised in downstream analyses. Regions attributed to recombination were detected using Gubbins[54]. These recombinogenic elements were removed, alongside previously determined mobile genetic elements, invariant sites, and columns with at least 2% indeterminate bases, to create a final alignment of 3,980 SNPs across 396 taxa (395 CenAsiaIII and one closely related outgroup). Maximum-likelihood phylogenies were constructed in five separate runs using RAxML v8.1.3 under the GTRCAT model, with sufficient bootstrap replicates automatically determined by the extended majority rule (MRE) bootstrap convergence[55]. Trees were visualised in association with metadata using the web-based interactive Tree of Life (iTOL)[56].

**Hierarchical Bayesian clustering for population structure**. Phylogeny-independent hierarchical Bayesian clustering (hierBAPS) was applied to the above alignment to assess population structure within the 395 CenAsiaIII isolates[57,58]. This was performed in ten independent analyses, with three layers of sub-clustering and the maximum number of clusters ranging from 10 to 40. These analyses identified two populations of identical membership in 8/10 instances, with all FQr *S. sonnei* belonging to a single population (named Pop2); all FQs belong to Pop1.

**Bayesian phylogenetic inference**. Aiming to examine the temporal structure of the CenAsiaIII clade, we subsampled to include 144 sequences (72 from each hierBAPS determined population, with a maximum of five isolates per calendar year in each geographic region). This subsampling generated an alignment of 2448 recombination-free SNPs, and a maximum-likelihood phylogeny was inferred as described above. A TVM (transversion) model of nucleotide substitution was determined to be the most appropriate for this alignment, using the fast ModelFinder implemented in IQ-TREE v1.6.7[59,60]. Path-O-Gen (v1.4) was used to assess the linearity of the relationship between root-to-tip divergence and sampling date (in month/year; $R^2 = 0.845$), and BEAST v.1.8.3 was used to estimate the substitution rate and time to the most recent common ancestor (tMRCA) of the CenAsiaIII clade[61]. To identify the best-fit model for the dataset, multiple BEAST runs were conducted utilizing combinations of the TVM + $\Gamma_4$ substitution model, strict or relaxed lognormal clock models, and constant or Bayesian skyline demographic models. Each of these analyses was performed in triplicate using a continuous 200 million generation MCMC chain with samples taken every 20,000 generations. Parameter convergence was assessed visually in Tracer v1.5 (ESS > 200). For robust model selection, both path sampling and stepping-stone sampling

approaches were applied to each BEAST analysis to estimate the marginal likelihood[62,63]. The best model selected by Bayes factor comparison was a strict molecular clock with a piecewise Bayesian skyline demographic model. For this model, triplicate runs were combined using LogCombiner v1.8.3, with removal of 20% burn-in. We additionally repeated the same procedure for 141 selected Pop2 *S. sonnei* to estimate the divergence time of the FQr clone. These analyses again indicated that the TVM + $\Gamma_4$ substitution model with a strict clock and Bayesian skyline demographic model was the best fit to the data. We selected a subset of 134 exclusively FQr *S. sonnei* (all containing *gyrA* S83L, *parC* S80I, and *gyrA* D87G) to infer the population dynamics of the clone since its emergence to avoid artificial interpretation of population expansion created by mixing multiple subpopulations[64]. BEAST runs were performed in triplicate and conducted using the TVM + $\Gamma_4$ substitution model, a strict molecular clock with the constant, Bayesian skyline, skygrid, or skyride demographic model[65,66]. The best-fit of these demographic models was the Bayesian skyline model.

**Phylogeographical analysis of CenAsiaIII *S. sonnei***. The reference-based maximum likelihood phylogeny of 395 CenAsiaIII *S. sonnei* served as an input for reconstructing the ancestral geographical states of each isolate, as adopted from a recently described approach[67]. We treated geographical source (sub-continental level) of the organisms as discrete characters. Due to biased sampling toward South Asia, we subsampled the phylogeny to include equal numbers of isolates from each character state (i.e. geographical region; $n = 13$ from South Asia, Africa, Europe, and combined Southeast Asia/East Asia), generating 1000 subsampled trees. Regions with fewer than 13 isolates (i.e. Middle East and America) were excluded. We utilised stochastic mapping to quantify transition events between geographical characters as well as the total time spent within each character state, separately for each subsampled tree[68]. Stochastic mapping, implemented using the function *make.simmap* in the R package phytools v0.6.0, was performed under an asymmetric model of character change (ARD) with the rate matrix sampled from the posterior probability distribution using MCMC (Q = mcmc) for 100 simulations[69]. To ensure that resultant phylogeographical signal was not an artefact due to oversampling in South Asia, we permuted (without replacement) the tip-location of the original maximum likelihood phylogeny, generating ten independent randomisation sets. We repeated stochastic mapping analysis for each randomisation as detailed above, using only 500 subsampled trees from each randomisation to reduce computational expense. Results from the 'true' and 'randomisation' runs were compared by analysis of variance (ANOVA) with post-hoc Tukey test.

**Determination of the accessory genome**. We performed a de novo sequence assembly using Velvet v1.2.03 and VelvetOptimizer for each isolate, and each read set was mapped back to the corresponding assembly[70]. Contigs with size <300 bp were eliminated, and annotation was determined for each assembly using Prokka[71]. ARIBA, a platform to detect genetic elements of choice from short reads, was deployed to detect the presence of antimicrobial resistance (AMR) genes, plasmid incompatibility types, and colicin types based on the reliably curated ResFinder, PlasmidFinder, and an in-house colicin database, respectively[72–74]. To identify contigs associated with the accessory genome, the assembly of each organism was ordered with reference sequences of Ss046 and the virulence plasmid (pSS046) using ABACAS. Contigs identified to belong to plasmids were queried against the public nucleotide databases using BLASTN. Artemis and Artemis Comparison Tool (ACT) were used to visualise the presence of specific genetic elements in the isolates[75].

**Detection of pseudogenes and gene loss**. Pseudogenes can be formed by numerous mechanisms in *Shigella*, including gene deletion, gene disruption by insertion sequences (IS) or small indels, and substitutions leading to gain of stop codons or loss of start codons. We selected an FQr CenAsiaIII isolate for long-read Nanopore sequencing (sample 2012–02037) to produce a high-quality complete reference for identification of potential pseudogenes. We compared the entire genome sequence of 2012–02037 to Ss046 reference using ACT to identify regions pertaining to genomic deletions or disrupted by IS elements, but discounting such events that occurred within IS elements. In total, we detected sixteen candidate events, including two genomic deletions (*aadA2* and *fliCAZY*) and 14 IS disruptions. The pan-genome of 395 CenAsiaIII *S. sonnei* sequences was then constructed using Roary with default setup[76], and gene losses were queried in all 395 isolates to record their presence/absence within the genome sequences. For the 14 IS disruption events, the intactness of these was queried against all CenAsiaIII strains using ARIBA, setting the matching length cutoff to 95%. This showed that nine such disruptions were stably inherited throughout the phylogeny (Supplementary Table 1).

To detect pseudogenes formed by substitutions and small indels, we extracted 393 CenAsiaIII variant calling format (vcf) files (two could not be retrieved) generated by mapping the reads to reference Ss046. Mutations were annotated with SnpEff v4.1b using the Ss046 reference, and were retained if they passed the quality filtering step and were classified by SnpSift v4.1b as (i) indels within genes, (ii) loss of a start codon, or (iii) gain of a stop codon[77]. Identified mutations within IS and repetitive elements were considered unreliable and removed from further analyses.

**dN/dS analysis on substitutions in CenAsiaIII *S. sonnei*.** For the 395 CenAsiaIII sequences, substitutions across the reference Ss046 were summarised and annotated using SnpEff v4.1b. Point mutations present within mobile, repetitive, or recombination elements (as defined in the 'phylogenetic reconstruction' section) were excluded, resulting in a set of 4230 mutations available for examination. Due to the limited temporal sampling period, we assessed the dN/dS ratio (non-synonymous to synonymous substitution rate) to detect potential signals of selection, adopted from a previously recommended approach[23,78]. In brief, a phylogenetic tree was constructed using these 4230 SNPs in RAxML v8.1.3, and PAML was used for ancestral state reconstruction of all SNPs[79]. Mutations were classified as intergenic, synonymous, or non-synonymous based on comparison of each SNP to its annotation and the reconstructed ancestral state. The dN/dS rate was adjusted for transition/transversion rate and codon frequencies under the NY98 model[80]. Genes were determined to be under positive selection if their adjusted dN/dS ratio was >1.5 or if they possessed no synonymous and at least five non-synonymous mutations (Table 2, Supplementary Table 3).

**Experimental evolution.** To examine how continuous exposure to FQ may impact the evolution of *S. sonnei*, we selected a representative isolate from each of the FQr (CenAsiaIII) and the FQs (GlobalIII) clades. Both isolates, cipR-VN (*gyrA*-S83L, *parC*-S80I, and *gyrA*-D87G) and cipS-VN (*gyrA*-S83L), were isolated in 2015 as part of a hospital-based diarrheal surveillance study conducted in Ho Chi Minh City, Vietnam[81]. Isolates were selected based on sequencing quality and the absence of plasmids encoding resistance to 3rd generation cephalosporins and macrolides (to discount potential interactions). The MICs to ciprofloxacin and nalidixic acid were reconfirmed by E-test and broth dilution. M9 minimal was used as culture medium for the experiment to limit over-proliferation. The M9 medium was supplemented with glucose as the sole carbon source (final concentration, 0.4%), and niacin (final concentration, 25 μg/mL) as *Shigella* cannot synthesise this vitamin de novo[82].

CipR-VN and cipS-VN were subjected to three longitudinal culture conditions in duplicate: (i) M9 (glucose + niacin) medium only, (ii) M9 (glucose + niacin) supplemented with 256 μg/mL of nalidixic acid, and (iii) M9 (glucose + niacin) supplemented with half the appropriate MIC of ciprofloxacin (0.125 μg/mL for cipS-VN and 4 μg/mL for cipR-VN). Cultures were incubated at 37 °C with continuous agitation for eight weeks, a duration expected to be sufficient to generate ciprofloxacin-resistant progenies from cipS-VN[83]; a sub-culture was performed every 24 h by transferring 20 μL into 5 mL of fresh medium. This process resulted in approximately eight bacterial generations ($\log_2(5000/20)$) per 24 h. At the end of the experiment (D56), organisms were harvested from 2 mL culture from each of the twelve experiments. Total genomic DNA was extracted from each sample using the Wizard Genomic DNA Extraction kit (Promega), and these were predicted to represent *S. sonnei* population diversity within each condition. Library preparation and deep sequencing by Illumina MiSeq (~100× coverage) were performed for all twelve samples to generate 2 × 250 bp paired-end reads. For each sample, reads were mapped to reference Ss046 by BWA-MEM, and high-quality SNPs were called by SAMtools and retained if they had a consensus quality > 50, mapping quality > 30, and read depth > 3[84]. Analyses were then performed independently for cipR-VN and cipS-VN, using existing sequence data as timepoint zero (D0). The Genome Analysis Toolkit (GATK) was employed to extract differing mutations between the paired sequences (D0 and D56), eliminating those identified in all conditions[85]. We additionally confirmed the fidelity of these mutations using Artemis and ACT. De novo assemblies were constructed for each isolate using SPAdes v3.9.0[86], and a pan-genome was constructed separately for cipR-VN and cipS-VN by Roary to identify genomic deletions.

**Reporting summary.** Further information on research design is available in the Nature Research Reporting Summary linked to this article.

## Data availability

Short-read sequence data generated for this study are available in the European Nucleotide Archive under project ERP012925 and runs ERR586829-ERR586845, ERR591140-ERR591218, and in the NCBI Sequence Read Archive under project PRJNA320210. The raw sequencing reads for the in vitro evolution experiment are available in NCBI Sequence Read Archive under project PRJNA553864. The phylogeny and associated metadata have been uploaded to Microreact for interactive viewing (https://microreact.org/project/ByXyXYgle). NCBI accession numbers for plasmids detected in our isolates are included in Supplementary Data 2. The source data underlying Fig. 3 and Supplementary Figs. 2 and 3 are available within the Source Data file. All other data supporting the findings of this study are available in this article and its Supplementary Information files.

## Code availability

Relevant source data and custom R codes used for stochastic mapping and dN/dS selection analysis described in the Methods section are deposited in Figshare (https://figshare.com/projects/Dissecting_the_molecular_evolution_of_fluoroquinolone_resistant_Shigella_sonnei/65840).

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

## Acknowledgements

H.C.T. received a DPhil scholarship from the Tropical Network Fund, Nuffield Department of Medicine, University of Oxford. S.B. is a Sir Henry Dale Fellow, jointly funded by the Wellcome Trust and the Royal Society (100087/Z/12/Z). We thank I. Carle, M. Lejay-Collin, and C. Ruckly from the Institut Pasteur for their excellent technical assistance. F.X. W is funded by the Institut Pasteur, Santé Publique France, and by the French Government "Investissement d'Avenir" program (Integrative Biology of Emerging Infectious Diseases Laboratory of Excellence, grant no. ANR-10-LABX-62-IBEID).

## Author contributions

H.C.T. and D.P.T. contributed to data analysis and interpretation of the results under the scientific guidance of M.A.R. and S.B.; H.C.T. and C.B. designed and performed the experimental evolution work. H.C.T. drafted and edited the paper, with D.P.T., M.A.R., and S.B. revising and structuring the paper. C.J., F.X.W., B.P.H., M.V., N.D.L., M.C., S.W., L.B., C.J.M., T.N.T.N., T.H.T., P.V.V., D.V.T., N.P.H.L., P.T., and P.J.C. contributed to sample collection, storage and DNA sequencing. R.W. and K.E.H. performed the long-read sequencing and assembly of FQr *S. sonnei*, using the Oxford Nanopore platform. N. R.T. provided access to sequencing facilities. G.T., K.E.H., and N.R.T. contributed to the editing of the paper. All authors read and approved the final draft.

## Competing interests

The authors declare no competing interests.
