## [Peer Review File · Nature Communications]

Reviewers' comments:

Reviewer #1 (Remarks to the Author):

The manuscript titled "Dissecting the molecular evolution of fluoroquinolone resistant *Shigella sonnei*" describes the location and timeline of the origin of the dominant fluoroquinolone resistant clone of *S. sonnei*. The authors used comparative genomics of a large global collection of *S. sonnei* to describe the genetic relatedness of fluoroquinolone resistant and sensitive isolates and to determine the location and timeline of emergence of the dominant fluoroquinolone resistant clone. The authors also performed in vitro evolution studies to examine differences in the mutational profiles of fluoroquinolone resistant and sensitive isolates following exposure to nalidixic acid and ciprofloxacin over eight weeks. Overall the manuscript provides important insight into the location of origin and mechanisms involved in the evolution of fluoroquinolone resistant *S. sonnei*.

Lines 164-167: Could you refer to the highlighting of these groups in the phylogeny more specifically in the text. It appears to be described in the Figure legend, but would be helpful to more quickly locate these groups when looking at the phylogeny if this is briefly referred to in the text.

Lines 205-207: Please clarify this sentence with respect to the numbers mentioned for the other species. Do these numbers reflect the number of related plasmids in public databases that exhibit similarity to the *S. sonnei* plasmids? This is not clear.

Lines 212-214: Was the determination that pSSE3 was maintained since 1995 based on detection of this plasmid in isolates as far back as 1995? Were there earlier isolates that did not have the plasmid?

Lines 286-288: Are the genes listed in Table S5 and described in this sentence all of the genes undergoing positive selection?

Lines 301 and 575-578: Is there a rationale for why the in vitro experiments were performed for eight weeks?

Table S5: Please describe in more detail the recombination region. Also, is there a reason why genes with ≥ 7 non-synonymous SNPs are listed in the table while genes with 5 or 6 non-synonymous SNPs are not listed?

Table S2: Please describe the table contents with footnotes or clarified column headers. For instance, do the isolate names belong to *S. sonnei* isolates? Do the descriptions of plasmids in column incB-O refer to previously described plasmids? If so, please provide the accessions of the named plasmids with the plasmid name. Does the species column indicate that the reference plasmids were all identified in *E. coli*? And under the Note column are these the differences in the *S. sonnei* plasmid compared to the reference plasmids? Why are some cells empty? Is it because the plasmids were identical to the references?

Fig S1: In the figure legend please describe the meaning of the branch colors.

Reviewer #2 (Remarks to the Author):

Of the four species of *Shigella*, *S. sonnei*, has been more commonly associated with shigellosis in developed countries. This species is now spreading across several countries in Latin America,

Africa, Middle East and Asia. The specific causes behind the global spread of *S. sonnei* are unclear, but genomic changes and selective pressure from localized antimicrobial use seems to have an impact on the microevolution of its population. In this study, the authors have used almost 400 *S. sonnei* isolates collected from endemic and non-endemic regions and performed whole genome sequence analysis to define the relationships of fluoroquinolone resistance with reference to in silico and in vitro functional aspects along with the evolutionary lineages in different populations.

Following are the comments.

1. What is missing in this investigation is inclusion *S. sonnei* isolates from the other potential endemic regions like Africa and India subcontinent. Does the global reference (14 isolates/sequences) cover these regions?

2. Does the isolates from diarrhoeal cases alone or inclusive of MSM, shigellemia etc?

3. Apart from the mutations in the QRDR regions, antibiotic resistance in *Shigella* spp. commonly occurs due to mobile genetic elements such as R-plasmids, transposons and integrons. Among, these, class-1 and class-2 integrons and drug resistance gene cassettes have been studied extensively. This aspect needs to be analysed and presented in the horizontal gene transfer section.

1. Sequence determination of the QRDR regions of GyrA at Gln69/Trp, Phe71/Ser, Ser72/Pro, Met75/Leu, Asp87Asn/Gly, Ser90/Cys, Met94/Leu, His106/Pro, Asn161/His, Thr163/Ala and in ParC at Ala64/Asp, Met86Trp, Ser129Pro reported in *S. sonnei* is considered in the sampling?

2. Fluoroquinolone resistance associated with azithromycin and third-generation cephalosporin resistance due to mphA and blaCTX alleles harbouring IncB/O/K/Z needs to be covered.

3. In several studies, role of plasmid-mediated quinolone resistance in *S. sonnei* has been described for (fluro)quinolone resistance. This aspect was not covered extensively in this study.

4. aac(6')-Ib gene encodes aminoglycoside acetyltransferase that is capable of acetylating the piperazinyl substituent of some fluoroquinolones and reduce their activities. This aspect is also important for the discussion.

5. Multiple-locus variable number tandem repeat analysis is being used for *S. sonnei* genotyping in epidemiological investigations. MLVA information on MDR isolates generated from the WGS analysis may carry a strong impact as well as comparing the overall phylogeny between MLA and SNP analysis.

6. MDR always correlates well with increased virulence in several pathogens. Shigellosis is associated with many virulence factors or interlinked with several genes. Comparing the association between virulence and AMR gene profiling can cover this aspect.

7. From the outbreaks of shigellosis in California, increased virulence was found to be associated with the presence of genes encoding Shiga toxin in fluoroquinolone resistant *S. sonnei* isolates (Kozyreva VK et al. 2016). It will be worth to identify a similar trend in other countries by analysing the stx in all the sequences.

8. Line 218-220. Identification of different colicin families is an interesting aspect. Does this correlate with the resistance profiles in Pop-1/Pop-II?

9. Line 98. 'Novel organisms' Change to fresh isolates.

Reviewers' comments:

Reviewer #1 (Remarks to the Author):

The manuscript titled “Dissecting the molecular evolution of fluoroquinolone resistant *Shigella sonnei*” describes the location and timeline of the origin of the dominant fluoroquinolone resistant clone of *S. sonnei*. The authors used comparative genomics of a large global collection of *S. sonnei* to describe the genetic relatedness of fluoroquinolone resistant and sensitive isolates and to determine the location and timeline of emergence of the dominant fluoroquinolone resistant clone. The authors also performed in vitro evolution studies to examine differences in the mutational profiles of fluoroquinolone resistant and sensitive isolates following exposure to nalidixic acid and ciprofloxacin over eight weeks. Overall the manuscript provides important insight into the location of origin and mechanisms involved in the evolution of fluoroquinolone resistant *S. sonnei*.

We thank the reviewer for the insightful comments and suggestions. The manuscript has been revised accordingly, as detailed below.

Lines 164-167: Could you refer to the highlighting of these groups in the phylogeny more specifically in the text. It appears to be described in the Figure legend, but would be helpful to more quickly locate these groups when looking at the phylogeny if this is briefly referred to in the text.

We have revised the manuscript as suggested. “(Vietnam, Thailand, and Cambodia; yellow box in Fig. 2)” and “(Ireland, Italy, Germany, and Spain; orange box in Fig. 2)”.

Lines 205-207: Please clarify this sentence with respect to the numbers mentioned for the other species. Do these numbers reflect the number of related plasmids in public databases that exhibit similarity to the *S. sonnei* plasmids? This is not clear.

We have revised the manuscript to clarify this, as followed. “The IncI1 plasmids, present in 42 *S. sonnei* isolates, displayed greater diversity than the IncB-O plasmids with respect to their origins. These IncI1 plasmid structures were similar to IncI1 plasmid structures found in GenBank for *Salmonella* spp. (n=14 plasmids), *E. coli* (n=13), *S. sonnei* (n=11), and *K. pneumoniae* (n=3) (Supplementary Table 2).”

Lines 212-214: Was the determination that pSSE3 was maintained since 1995 based on detection of this plasmid in isolates as far back as 1995? Were there earlier isolates that did not have the plasmid?

The timing for the acquisition of pSSE3 was inferred by Bayesian phylogenetic analysis (using BEAST), which was defined as the time to most recent common ancestor (tMRCA) of the phylogenetic clade harbouring the pSSE3 (Fig 2, Fig S1). This tMRCA coincided with the inferred time of the first *gyrA* mutation S83L (Fig 2). Since pSSE3 was detected in the majority of isolates after this inferred tMRCA, we speculated that the plasmid was introduced once in the *S. sonnei* population circa 1995 and was stably inherited. Additionally, we observed 5 *S. sonnei* strains, isolated from Thailand in 1994, that did not carry the plasmid. These strains were outgroups to the pSSE3-harboring clade. We have edited the manuscript for clarification. “Through genomic comparison and Bayesian phylogenetic inference, we confirmed that this plasmid has been stably maintained in this clade since its introduction around the year 1995, concurrent with the first *gyrA* S83L mutation (Supplementary Fig. 1).”

Lines 286-288: Are the genes listed in Table S5 and described in this sentence all of the genes undergoing positive selection?

Yes. The 14 genes described in Table S5 are predicted to be undergoing positive selection. These were the output from examination of 3,637 genes of *S. sonnei*. We have added in the manuscript for clarification as follows: "Among the 3,637 *S. sonnei* homologs examined, 14 genes were predicted to have undergone positive selection (adjusted dN/dS ranging from 1.8 to 3.6) (Supplementary Table 5)".

Lines 301 and 575-578: Is there a rationale for why the in vitro experiments were performed for eight weeks?

For the *in vitro* experiment, we adapted the methods used in a previous study (Long et al., 2016, PNAS), which showed measurable degrees of mutation accumulation across the genome in sensitive *E. coli* exposed to sublethal concentrations of fluoroquinolone for ~1-2 months, with the rate of mutation accumulation enhanced by exposure. Thus, we hypothesized that 8 weeks of exposure to fluoroquinolone was sufficient to generate ciprofloxacin-resistant progeny from the Global III *S. sonnei*. We have added in the Methods for clarification. "Cultures were incubated at 37°C with continuous agitation for eight weeks, a duration expected to be sufficient to generate ciprofloxacin resistant progenies from cipS-VN (Long et al., 2016, PNAS)."

Table S5: Please describe in more detail the recombination region. Also, is there a reason why genes with ≥7 non-synonymous SNPs are listed in the table while genes with 5 or 6 non-synonymous SNPs are not listed?

In Table S5, we also indicated the number of genes with varying numbers of mutations. This showed that while 6 genes have 5 non-synonymous mutations, 20 genes have 4 non-synonymous mutations. Thus, a fixed-cutoff of 5 non-synonymous mutations is included when genes have no synonymous mutations (as written in the Methods section). Table S5 includes genes with 5 or more non-synonymous mutations. The recombination region for 395 CenAsialIII *S. sonnei* alignment is as defined in the phylogenetic reconstruction section in the Methods. Mutations falling within the identified recombination region are not suitable for selection analysis, as they do not represent inheritable changes in homologs. We have amended the manuscript to read "Point mutations present within mobile, repetitive, or recombination elements (as defined in the 'phylogenetic reconstruction' section) were excluded, resulting in a set of 4,230 mutations available for examination", and have added information to Table S5.

Table S2: Please describe the table contents with footnotes or clarified column headers. For instance, do the isolate names belong to *S. sonnei* isolates? Do the descriptions of plasmids in column incB-O refer to previously described plasmids? If so, please provide the accessions of the named plasmids with the plasmid name. Does the species column indicate that the reference plasmids were all identified in *E. coli*? And under the Note column are these the differences in the *S. sonnei* plasmid compared to the reference plasmids? Why are some cells empty? Is it because the plasmids were identical to the references?

We have revised Table S2 for clarity and added a new worksheet called 'Footnote' to explain the column names in detail. The detected incB-O plasmids in our *S. sonnei* strains are either closely related to pHUSEC41-1 (>98% nucleotide identity) or its recombinants (with pEB1 and pFORC11.2). These are described in the original manuscript. We have provided the accession numbers of these plasmids and

other frequently detected plasmids in our samples in the “Footnote” worksheet. The information in the ‘Note’ column details additional information, and its absence in some cells means that there is no additional information relevant to the study to be considered. Since the plasmids were assembled from short read sequencing and are not complete sequences, we could not exhaustively assess their identities (structure-wise) relative to the references.

Fig S1: In the figure legend please describe the meaning of the branch colors.

We have edited Fig S1 legend to include “The branch colour scheme indicates bootstrap support for the corresponding branch, ranging from low to high (red to black).”

Reviewer #2 (Remarks to the Author):

Of the four species of *Shigella*, *S. sonnei*, has been more commonly associated with shigellosis in developed countries. This species is now spreading across several countries in Latin America, Africa, Middle East and Asia. The specific causes behind the global spread of *S. sonnei* are unclear, but genomic changes and selective pressure from localized antimicrobial use seems to have an impact on the microevolution of its population. In this study, the authors have used almost 400 *S. sonnei* isolates collected from endemic and non-endemic regions and performed whole genome sequence analysis to define the relationships of fluoroquinolone resistance with reference to in silico and in vitro functional aspects along with the evolutionary lineages in different populations.

We thank the reviewer for his/her constructive comments. We have addressed the reviewer’s questions and revised the manuscript where appropriate. We hope this revised version adds more clarity to the study.

Following are the comments.

1. What is missing in this investigation is inclusion *S. sonnei* isolates from the other potential endemic regions like Africa and India subcontinent. Does the global reference (14 isolates/sequences) cover these regions?

As the time this study commenced, epidemiological reporting of fluoroquinolone resistant *S. sonnei* from Africa was limited. Recent studies highlight that the major burden of shigellosis in African countries is caused by *S. flexneri* (Livio et al., 2014, Clinical Infectious Diseases). Thus, it is likely that Africa is not currently a major reservoir for fluoroquinolone resistant *S. sonnei*. The public health laboratories in non-endemic settings include samples collected from patients with travel history, including those traveling from Africa and the Indian subcontinent (South Asia). There were 13 African *S. sonnei* isolates included in our dataset (Table S1 and Fig 1), of which 8 were ciprofloxacin resistant and 3 were from global references. The majority of resistant isolates originated from South Asia (Table 1, Fig1, FigS1), including organisms from India, Nepal, Bhutan, Bangladesh. Though we did not directly collect *S. sonnei* from diarrhoeal surveillance in South Asia, we predict the isolates included here to be largely representative of the circulation of *S. sonnei* (both resistant and susceptible to FQ) in this geography.

2. Does the isolates from diarrhoeal cases alone or inclusive of MSM, shigellemia etc?

All *S. sonnei* included in this study were isolated from diarrhoeal stool samples. For *S. sonnei* isolated from adults in non-endemic settings, we did not have additional metadata to address the circulation of fluoroquinolone resistant *S. sonnei* in the MSM community. There is a recent paper describing the circulation of FQ resistant *S. sonnei* in MSM in the UK (K. Baker et al., 2018, Nature Communications), and these organisms belong to the CenAsialIII expansion described in this study. This has been mentioned in the Discussion.

3. Apart from the mutations in the QRDR regions, antibiotic resistance in *Shigella* spp. commonly occurs due to mobile genetic elements such as R-plasmids, transposons and integrons. Among, these, class-1 and class-2 integrons and drug resistance gene cassettes have been studied extensively. This aspect needs to be analysed and presented in the horizontal gene transfer section.

Class II integron in *S. sonnei* has been described in previous studies, conferring resistance to trimethoprim *dfrA1*. We did include *dfrA1* in the analysis but did not state specifically. We have edited the manuscript as suggested. "Genes encoding resistance to previous first-line antimicrobials for treating shigellosis were commonplace within the clade, as observed previously^{10,11}. These included the small plasmid *spA*, conferring resistance to streptomycin (*strAB*), tetracycline (*tetRA*) and sulphonamide (*sul2*), and a class II integron conferring resistance to trimethoprim (*dfrA1*)."

1. Sequence determination of the QRDR regions of GyrA at Gln69/Trp, Phe71/Ser, Ser72/Pro, Met75/Leu, Asp87Asn/Gly, Ser90/Cys, Met94/Leu, His106/Pro, Asn161/His, Thr163/Ala and in ParC at Ala64/Asp, Met86Trp, Ser129Pro reported in *S. sonnei* is considered in the sampling?

For the sampling, we selected the ciprofloxacin resistant isolates based on results from antimicrobial susceptibility testing, regardless of the genetic mechanisms underlying the resistance. We followed the reviewer's suggestion and examined the other mutations in GyrA and ParC of CenAsialIII *S. sonnei*, but we did not find any such mutations in our samples. The triple mutations that we described in GyrA and ParC are known to increase the ciprofloxacin MIC 60-fold (Redgrave et al., 2014, Trends in Microbiology), and they are the main QRDR mutations in clinical Enterobacteriaceae isolates.

2. Fluoroquinolone resistance associated with azithromycin and third-generation cephalosporin resistance due to *mphA* and *blaCTX* alleles harbouring *IncB/O/K/Z* needs to be covered.

The co-transfer of *mphA* and ESBL in a fluoroquinolone resistance background is of high interest in Vietnam. We have characterized these *incB-O* plasmids in greater detail in a separate publication, as this was too much work to additionally incorporate in this study. Notably, we used long read sequencing to characterize the complete sequence of this plasmid. This manuscript is currently under revision at [redacted] and is available to the reviewer and handling editor on request [redacted]. We will cite this paper once published.

3. In several studies, role of plasmid-mediated quinolone resistance in *S. sonnei* has been described for (fluro)quinolone resistance. This aspect was not covered extensively in this study.

We searched the *S. sonnei* genomes for all known acquired antimicrobial resistance genes. *qnrS1* is only found in two *S. sonnei* isolates from South Asia, which already harboured the triple mutation. We did not find any isolates carrying other plasmid-mediated quinolone resistance genes such as *aac(6')-Ib-cr*, *qepA*, *oxqAB*. We have edited in the Results section to reflect this. "Plasmid-mediated quinolone resistance (PMQR) genes were uncommon in our isolates, with *qnrS1* found in two *S. sonnei* (already

harbouring the aforementioned triple mutation) and no organisms possessing *aac(6′)-Ib-cr*, *qepA* or *oxqAB*.”

4. *aac(6′)-Ib* gene encodes aminoglycoside acetyltransferase that is capable of acetylating the piperazinyl substituent of some fluoroquinolones and reduce their activities. This aspect is also important for the discussion.

See above.

5. Multiple-locus variable number tandem repeat analysis is being used for *S. sonnei* genotyping in epidemiological investigations. MLVA information on MDR isolates generated from the WGS analysis may carry a strong impact as well as comparing the overall phylogeny between MLA and SNP analysis.

MLVA is an outdated technique that provides insufficient resolution for the international tracking of clades and does not provide robust phylogenetic data. Genome sequencing allows for much greater resolution in characterizing these organism and is the gold standard, performing MLVA analysis on these samples would not provide any further valuable data in the context of this scientific investigation.

6. MDR always correlates well with increased virulence in several pathogens. Shigellosis is associated with many virulence factors or interlinked with several genes. Comparing the association between virulence and AMR gene profiling can cover this aspect.

Most virulence factors in *Shigella sonnei* are located on the large virulence plasmid (including the Type 3 Secretion system, Enterotoxin ShET2 and O-antigen), which is frequently lost upon laboratory subculture and storage. This phenomenon is well known in *Shigella* research (McVicker and Tang, 2016, Nature Microbiology). Therefore, the genomic data obtained from *S. sonnei* sequencing in this study is not suitable for examining the variability in virulence factors. Further, given the clonality and short evolutionary timeframe of CenAsiaIII *S. sonnei*, it is unlikely that the rise of FQ resistance has been coupled with acquisition/deletion of virulence factors. Our recent clinical study “demonstrated that disease severity and recovery in *S. sonnei* infected children were independent of the organism’s ciprofloxacin susceptibility profile (Duong Vu Thuy et al., 2017, Clinical Infectious Diseases)” (Discussion), suggesting that resistance in *S. sonnei* is not likely to be associated with increased virulence.

7. From the outbreaks of shigellosis in California, increased virulence was found to be associated with the presence of genes encoding Shiga toxin in fluoroquinolone resistant *S. sonnei* isolates (Kozyreva VK et al. 2016). It will be worth to identify a similar trend in other countries by analysing the *stx* in all the sequences.

In the paper mentioned by the reviewer (Kozyreva et al., 2016), two distinct outbreaks were observed in California. One was associated with fluoroquinolone resistance and belonged to the CenAsiaIII expansion, the other carried the Shiga toxin gene (*stx*) and belonged to the Global III clade. Therefore, presence of *stx* variants has not been described in fluoroquinolone resistant *S. sonnei*. In our study, through pan-genome examination and *stx* sequence search, we did not observe the presence of *stx* variants in any isolates.

8. Line 218-220. Identification of different colicin families is an interesting aspect. Does this correlate with the resistance profiles in Pop-1/Pop-II?

It is somewhat unclear which resistance profile the reviewer is referring to in this case. Colicin is a narrow-spectrum bacterial toxin, and resistance to colicin is usually attributed to mutations in the cobalamin transporter *btuB* (non-specific) or possession of colicin immunity genes (specific). However, we did not investigate colicin resistance in our *S. sonnei* isolates as it is not relevant to this manuscript's aim. If the reviewer is referring to differences in antimicrobial resistance profiles between Pop-1 and Pop-2, we found no correlation between carriage of different colicin families and antimicrobial resistance profiles.

9. Line 98. 'Novel organisms' Change to fresh isolates.

We have edited this phrase to read 'contemporary isolates'.

REVIEWERS' COMMENTS:

Reviewer #1 (Remarks to the Author):

[No further comments for authors]

Reviewer #2 (Remarks to the Author):

2. Fluoroquinolone resistance associated with azithromycin and third-generation cephalosporin resistance due to mphA and blaCTX alleles harbouring IncB/O/K/Z needs to be covered.

The co-transfer of mphA and ESBL in a fluoroquinolone resistance background is of high interest in Vietnam. We have characterized these incB-O plasmids in greater detail in a separate publication, as this was too much work to additionally incorporate in this study. Notably, we used long read sequencing to characterize the complete sequence of this plasmid. This manuscript is currently under revision at [Redacted] and is available to the reviewer and handling editor on request [Redacted]. We will cite this paper once published.

Not sure which article will be published first. Better to mention "unpublished observation" or cite the reference if the other article get published before the current one.

RESPONSE TO REVIEWERS' COMMENTS:

Reviewer #1 (Remarks to the Author):

[No further comments for authors]

Reviewer #2 (Remarks to the Author):

2. Fluoroquinolone resistance associated with azithromycin and third-generation cephalosporin resistance due to mphA and blaCTX alleles harbouring IncB/O/K/Z needs to be covered.

The co-transfer of mphA and ESBL in a fluoroquinolone resistance background is of high interest in Vietnam. We have characterized these incB-O plasmids in greater detail in a separate publication, as this was too much work to additionally incorporate in this study. Notably, we used long read sequencing to characterize the complete sequence of this plasmid. This manuscript is currently under revision at [Redacted] and is available to the reviewer and handling editor on request [Redacted]. We will cite this paper once published.

Not sure which article will be published first. Better to mention "unpublished observation" or cite the reference if the other article get published before the current one.

As suggested by the editor of this paper and the editor of the [Redacted] paper, we have submitted the [Redacted] submission to bioRxiv and have now cited it in the manuscript with limited discussion (page 9, lines 228-229; citation #13).